

# A small molecule targeting protein translation does not rescue spatial learning and memory deficits in the hAPP-J20 mouse model of Alzheimer's disease

Erik C.B. Johnson[1,2] and  Jing Kang[1]

[1] Gladstone Institute of Neurological Disease, Gladstone Institutes, San Francisco, CA, United States
[2] Department of Neurology, University of California –San Francisco, San Francisco, CA, United States

## ABSTRACT

A small molecule named ISRIB has recently been described to enhance memory in rodents. In this study we aimed to test whether ISRIB would reverse learning and memory deficits in the J20 mouse model of human amyloid precursor protein (hAPP) overexpression, a model that simulates many aspects of Alzheimer's disease in which memory deficits are a hallmark feature. We did not observe a significant rescue effect with ISRIB treatment on spatial learning and memory as assessed in the Morris water maze in J20 mice. We also did not observe a significant enhancement of spatial learning or memory in nontransgenic mice with ISRIB treatment, although a trend emerged for memory enhancement in one cohort of mice. Future preclinical studies with ISRIB would benefit from additional robust markers of target engagement in the brain.

Corresponding author
Erik C.B. Johnson,
erik.johnson@gladstone.ucsf.edu

## INTRODUCTION

The prevalence of Alzheimer's disease (AD) is increasing as the human population ages, and treatments that slow or reverse the disease are urgently  needed (*Alzheimer's Association, 2015*). In AD, loss of the ability to form new memories, and eventually to recall long-term memories, is a defining clinical feature. While the causes of AD remain a focus of intense investigation, treatments that can enhance the brain's innate ability to form and retrieve memories, or that counteract the mechanisms the lead to memory loss, could offer an immediate benefit to the millions of people who currently suffer from this disease.

Recently, a symmetric bis-glycolamide named ISRIB (integrated stress response inhibitor) has been described to enhance spatial and fear memory in rodents when delivered immediately after a stimulus or completion of a behavioral task (*Sidrauski et al., 2013*). ISRIB binds to and activates the guanine nucleotide exchange factor eIF2B (elongation initiation factor 2 B), which in turn relieves inhibition of protein translation caused by phosphorylation of the alpha subunit of initiation factor 2 (eIF2$\alpha$) (*Sidrauski et al., 2013*; *Sekine et al., 2015*; *Sidrauski et al., 2015a*; *Sidrauski et al., 2015b*). While phosphorylation

of eIF2$\alpha$ is a key event in the integrated stress response (ISR), in which a diverse array of cellular stressors can lead to an overall decrease in the rate of protein translation (*Walter & Ron, 2011*; *Halliday & Mallucci, 2015*), baseline phosphorylation levels of eIF2$\alpha$ also provide a homeostatic "brake" on global protein translation rates even in the absence of cellular stress and activation of the ISR. Release of this brake through pharmacological activation of eIF2B by ISRIB has been proposed to lead to enhanced memory formation, possibly through enhancement of cAMP responsive element-binding protein (CREB) expression (*Costa-Mattioli et al., 2007*; *Stern et al., 2013*), a protein essential for memory formation (*Kida & Serita, 2014*), or through other mechanisms such as modulation of synaptic long-term depression (LTD) (*Di Prisco et al., 2014*).

Mouse models in which the human amyloid precursor protein (hAPP) is overexpressed in the brain have been widely used as models of AD (*Webster et al., 2014*). The hAPP-J20 (J20) mouse model contains the Swedish and Indiana hAPP mutations that cause familial AD, leading to overproduction of total levels of amyloid-beta (A$\beta$) as well as an increase in the A$\beta$1-42/A$\beta$1-40 ratio (*Mucke et al., 2000*). The J20 model recapitulates many aspects of AD, including the formation of aggregated A$\beta$ in the brain and the development of deficits in learning and memory (*Mucke et al., 2000*; *Webster et al., 2014*). Spatial learning and memory, as assessed by performance in the Morris water maze (*Morris, 1984*), is robustly impaired in J20 (*Roberson et al., 2007*; *Sanchez-Mejia et al., 2008*; *Cisse et al., 2011*; *Sanchez et al., 2012*). The precise mechanism(s) underlying this impairment remains under investigation, but one such mechanism may be enhanced LTD. LTD has been shown to be pathologically elevated in the hippocampus in the presence of aggregated forms of A$\beta$ (*Shankar et al., 2008*; *Luscher & Huber, 2010*; *Pozueta, Lefort & Shelanski, 2013*). Treatment with ISRIB prevents the transcriptional program required for induction of LTD (*Sidrauski et al., 2013*; *Di Prisco et al., 2014*), and may thereby provide an effective treatment for spatial learning and memory problems in both the J20 model and in patients with AD.

In this preclinical study, we tested whether ISRIB could enhance not only spatial learning and memory and fear memory in nontransgenic mice through a mechanism that remains to be completely defined (*Sidrauski et al., 2013*; *Stern et al., 2013*), but also whether it similarly could improve spatial learning and memory deficits in the J20 AD mouse model. In our experimental design, we made an effort to replicate as closely as possible the dosing regimens and behavioral protocols used in the prior study describing enhanced memory performance with ISRIB treatment (*Sidrauski et al., 2013*).

## MATERIALS AND METHODS

### Mice

hAPP-J20 mice were generated in-house and maintained on a C57BL/6J genetic background (*Mucke et al., 2000*). Mouse pups were weaned 4–6 weeks after birth and housed up to 5 per cage. Mice were fed a regular chow diet (PicoLab Rodent Diet 5053, TestDiet) and maintained on a 12-hour light/dark cycle. All animal experiments were approved by the Institutional Animal Care and Use Committee at University of California-San Francisco under protocol AN105527-03.

## Compounds and reagents

All chemicals were purchased from Sigma. *trans*-ISRIB was provided by Dr. Peter Walter (University of California-San Francisco). ISRIB was dissolved in DMSO, then diluted 100-fold in sterile-filtered 0.9% saline pH 7.0 to a final concentration of 15 µg/mL (0.25 mpk) or 6 µg/mL (0.1 mpk). For the 2.5 mpk dose, ISRIB was dissolved in 1:1 DMSO/PEG$_{400}$ at 0.75 mg/mL and administered at 3.3 µL/g. Treatment solutions were prepared fresh daily from the appropriate DMSO stock solution.

## Western blot analysis

Mice were anesthetized with 2,2,2-Tribromoethanol (Avertin), perfused with 0.9% saline, and their hemibrains removed and snap frozen on dry ice and stored at −80 °C. Prior to dissection, the hemibrains were thawed for 1 hour on ice, then dentate gyrus and cortex dissected and refrozen on dry ice. Samples were then thawed and homogenized on a magnetic bead homogenizer (NextAdvance Storm 24), followed by sonication (Episonic), in homogenization buffer (1× PBS, 1mM DTT, 0.5mM EDTA, 0.5% Triton X-100, 0.1M PMSF, 1× HALT$^{TM}$ protease/phosphatase inhibitor) at 4 °C. Homogenates were then centrifuged to pellet insoluble material, and the supernatants used for analysis. Protein concentrations were determined by the Bradford assay (Bio-rad) (*Bradford, 1976*). Equal amounts of protein (15 µg) prepared in 1× NuPAGE LDS sample buffer (Life Technologies) and 1x sample reducing buffer (Life Technologies) were loaded into the running lanes and electrophoresed at 180V for 2 hours on 4–12% Bis-Tris (Life Technologies) gels using a 1× MOPS buffer. Gels were soaked in 2× transfer buffer (Life Technologies) + 10% methanol for 20 mins, then proteins transferred to a nitrocellulose membrane using a dry transfer system (iBlot, Life Technologies). Membranes were blocked with either 5% BSA in Tris-buffered saline (TBS) (eIF2α) or with Odyssey blocking buffer (LI-COR) (ATF4) for 1 hour at room temp, then washed with TBS + 0.1% Tween 20 (TBST) three times for 5 mins each. Membranes were incubated with primary antibody in 5% BSA-TBST overnight at 4 °C, washed three times with TBST, then incubated with matching secondary antibody conjugated to IRDye (LI-COR) in Odyssey blocking buffer (LI-COR) + 0.2% Tween 20 for 1 hour at room temp. Membranes were washed three times with TBST, then imaged on an Odyssey CLx infrared imaging system (LI-COR).

## Antibodies and dilutions used for western blotting

| | | |
|---|---|---|
| ATF4 | (Santa Cruz) | 1:200 |
| GAPDH | (Millipore) | 1:1000 |
| IRDye 2° | (LI-COR) | 1:10,000 |

## Behavioral testing

Mice were group-housed during all behavioral tests. All control mice were littermate controls. The investigator performing the behavioral testing was blinded to genotype and treatment. The mice were handled prior to the start of behavioral testing. Five days prior to the start of the experiment the experimenter placed his hand in the cage for 5 min. On the second through fourth days of handling the experimenter placed his hand in the cage

for 1 min, then transferred the mice from the home cage to a clean cage for 2 min. On the day prior to the start of the experiment the mice were transferred to the experimental room and allowed to acclimate for 30 min, then handled for 1 min and transferred from the home cage to a clean cage for 2 min.

## Morris water maze

The water maze consisted of a pool (122 cm diameter) filled with water opacified with nontoxic white tempera paint powder, and surrounded by extramaze cues. For the pilot MWM #1, mice were trained to find a hidden platform (14 × 14 cm, submerged 1 cm) over 6 trials. For MWM #2, mice were trained over 11 trials. The platform location remained the same throughout hidden platform training, but the drop location varied semi-randomly between trials. Each trial was 120 s in length, and was performed once per day except for the last two days for MWM #2, in which 2 trials were performed per day (trials 8 and 9, and 10 and 11 were performed on the same day). The trial ended when the mouse found the platform, but the mouse was required to remain on the platform for 10 s prior to removal from the pool. After removal from the pool, the mouse was immediately injected with ISRIB by the intraperitoneal (i.p.) route, then placed back into the home cage, as previously described (*Sidrauski et al., 2013*). Memory for the location of the hidden platform was tested with 60-second spatial probe trials performed 24 and 72 hours after the final training trial. The drop location was 180° from where the platform was placed during hidden platform training. The same drop location was used for both spatial probe trials. Swim paths were recorded and analyzed using an Ethovision XT video tracking system (Noldus Information Technology). Swim speeds were not different between experimental groups. Training performance measures included latency to find the platform and distance traveled to find the platform. Probe performance measures included percent time spent in the target quadrant, latency to first cross the platform location, and the number of platform crossings.

## Contextual fear conditioning

Mice were placed in near infrared fear conditioning chambers (Med Associates Inc.) that contained distinct tactile, visual (variable light intensity set at 3), and olfactory (1% acetic acid solution) cues. The mice were allowed to explore the chamber for 2 min, at which time a 0.35 mA foot shock was delivered for 1 s. The activity of the mice was recorded for 1 min post-shock, after which the mice were removed from the chambers and immediately treated with ISRIB, as previously described (*Sidrauski et al., 2013*). Twenty-four hours later the mice were placed back into the same chambers and movements were observed for 8 min. Movements were recorded using high speed monochrome digital video cameras and analyzed with VideoFreeze software (Med Associates Inc.). Chambers were cleaned with 70% ethanol between each testing session.

## Statistical analysis

Statistical analysis was performed using Prism 6 (GraphPad Software) or the statistical programming language R (http://www.R-project.org/). Normality was assessed using the D'Agostino & Pearson omnibus test. Variance between groups was assessed using the *F*

test. For comparisons of normal distributions with equal variances, a two-tailed unpaired $t$ test was used. For comparisons of distributions in which one or both deviated from normality, a two-tailed Mann–Whitney test was used. Quadrant preference in the Morris water maze was assessed using a two-tailed one-sample $t$ test against a theoretical mean of 25%. Treatment effects in the pilot MWM probe were analyzed using a one-way ANOVA. Treatment effects in the J20 MWM probe were analyzed using a two-way ANOVA with Sidak's test for multiple comparisons. Morris water maze training data were analyzed using a linear mixed effects regression model for censored responses (*Vaida, Fitzgerald & Degruttola, 2007*) implemented in the R package lmec (*Vaida & Liu, 2015*). Gender and ISRIB were included as trial modifiers. Gender was removed from the final model after no significant effect was observed. Random mouse-level intercepts and slopes accounted for the correlation among repeated observations. Goodness of fit was analyzed by inspection of residuals. Differences were considered significant if the 5–95% confidence interval did not cross zero. Values reported are mean $\pm$ standard error of the mean. Differences were considered significant at $p < 0.05$.

## RESULTS

### A marker of ISRIB target engagement cannot be observed in brain homogenate from nontransgenic or hAPP-J20 mice

The transcription factor ATF4 has been used as a marker of target engagement in previously published preclinical trials of ISRIB in disease models in which the level of this protein is elevated (*Halliday et al., 2015*; *Palam et al., 2015*). In order to determine whether we could also use levels of this protein as a marker of ISRIB target engagement in the brain, we analyzed brain homogenates from two separate brain regions—cortex and dentate gyrus—in nontransgenic and hAPP-J20 mice at three different ages: 2–3, 6–7, and 12–13 months. Consistent with previous studies, we were unable to detect protein levels of ATF4 in nontransgenic mice by western blotting in either brain region (Fig. 1A) (*Halliday et al., 2015*). We were also unable to detect ATF4 in hAPP-J20 mice in these same brain regions. We noted a very weak band at approximately 50kD in the brain homogenates that appeared to be a nonspecific background band, but we nevertheless quantified this band on the possibility that it represented a uniquely post-translationally modified ATF4. The levels of this protein were not different between NTG and J20 mice in either cortex or dentate gyrus in all 3 age groups (Fig. 1B; unpaired $t$ test, $p > 0.05$). These results suggest that the ISR is not significantly elevated in hAPP-J20 mice or in nontransgenic mice. Therefore, we did not have a protein marker of ISRIB target engagement in the brain for this preclinical drug study, similar to previous studies on memory enhancement with ISRIB (*Sidrauski et al., 2013*; *Di Prisco et al., 2014*).

### In a pilot behavioral experiment, ISRIB showed a trend towards enhancement of spatial memory in the Morris water maze

Although we did not have a marker of target engagement in the brain for ISRIB, we decided to proceed with a pilot study to test its effects on learning and memory in nontransgenic mice. We used the Morris water maze as a test of spatial learning and memory

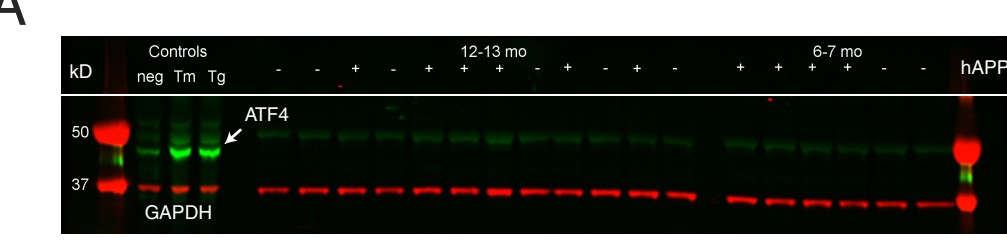

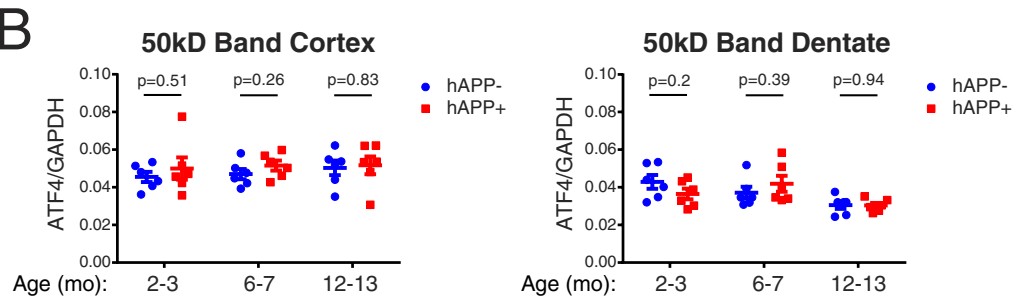

**Figure 1** **ATF4 as a pharmacodynamic marker in nontransgenic and hAPP-J20 mice.** (A) Cortex and dentate gyrus brain homogenates from hAPP-J20 (J20) and nontransgenic (NTG) mice at 2–3, 6–7, and 12–13 months (mo) of age were analyzed for ATF4 protein levels by western blotting ($N = 6$ per group). A representative blot is shown. While low levels of ATF4 could be observed in the untreated control 293T cell lysate, and elevated levels could be observed in the 293T cells treated with tunicamycin (Tm) and thapsigargin (Tg) to induce ER stress, ATF4 could not be observed in NTG or J20 brain homogenate. (B) A weak band at 50kD in the brain homogenate samples was quantified on the possibility that this represented a post-translationally modified ATF4 unique to brain tissue. No differences in protein levels represented by this band were observed between NTG and J20 mice (unpaired $t$ test, $p > 0.05$).

(*Morris, 1984*). We treated mice with 0.1 mg/kg (mpk) or 0.25 mpk ISRIB by intraperitoneal (i.p.) injection immediately after finishing each training session in the Morris water maze (MWM). We used a weak training protocol of one trial per day in order to increase our chances of observing an enhancement in learning and/or memory, as previously described (*Sidrauski et al., 2013*). Twenty-four and seventy-two hours after completion of the last training trial, we tested memory for the location of the hidden platform in probe trials. We did not observe a difference between control or ISRIB-treated mice at either dose in the time required to locate the hidden platform during training, or the distance traveled before locating the hidden platform (linear mixed effects regression models, non-significant 5–95% confidence intervals) (Fig. 2A). In the 24-hour probe trial, we did not observe a significant difference between control and ISRIB-treated mice in 3 separate outcome measures: percent time spent in the target quadrant (one-way ANOVA, $F_{(2,42)} = 0.45$, $p = 0.64$), latency to first cross the platform location (one-way ANOVA, $F_{(2,39)} = 1.18$, $p = 0.32$), or the number of platform crossings (one-way ANOVA, $F_{(2,42)} = 0.93$, $p = 0.40$) (Fig. 2B). However, there was a dose-related numerical trend towards improved performance in the latency measure with ISRIB treatment. At 72 hours, the mice treated with 0.25 mpk ISRIB spent significantly more time in the target quadrant compared to chance, whereas the control and 0.1 mpk ISRIB-treated mice did not

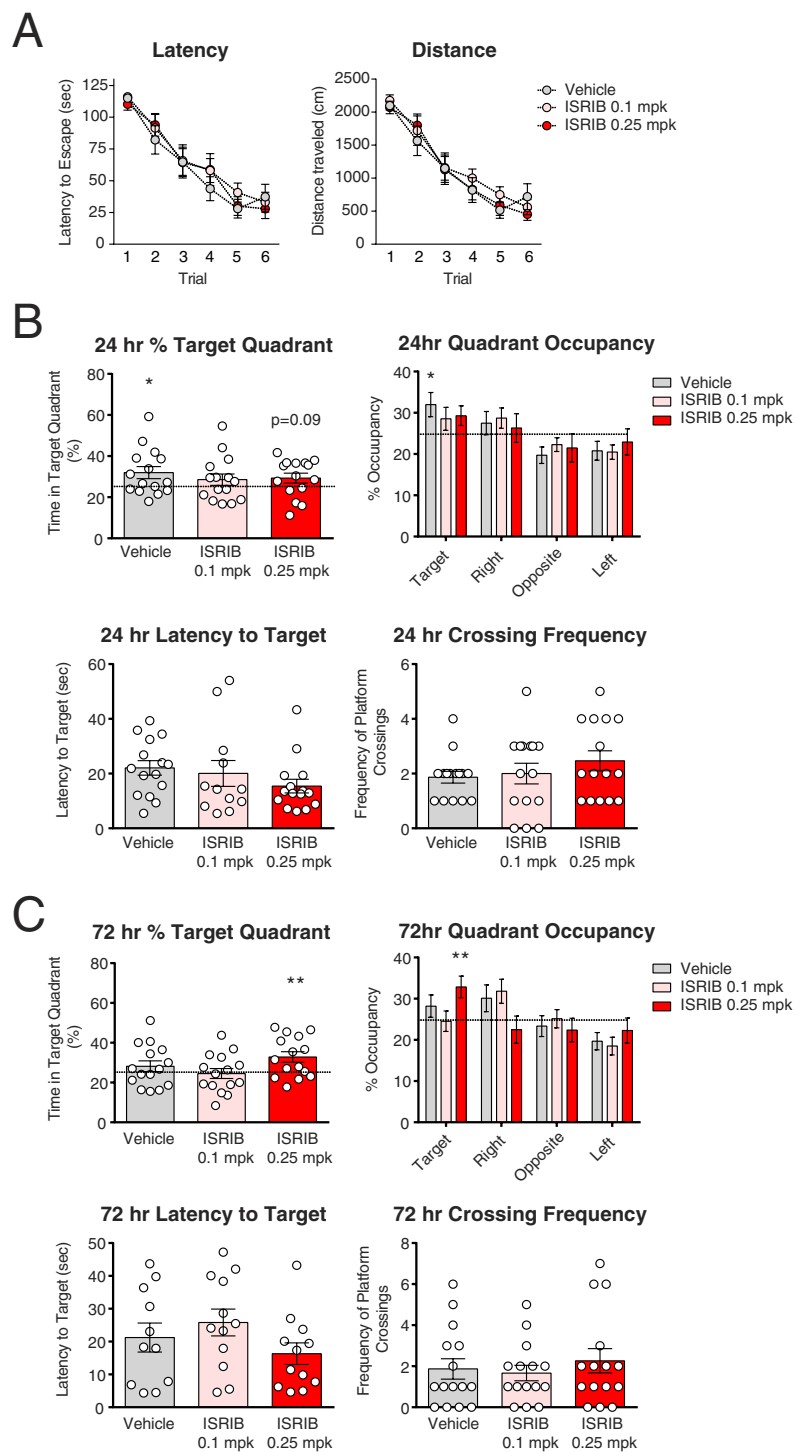

**Figure 2** **A pilot Morris water maze experiment suggests a trend towards enhanced long-term spatial memory with ISRIB treatment.** (A–C) 6–7 month-old nontransgenic mice were tested in the Morris water maze (MWM) ($N = 15$ per group). (A) MWM training. Mice were treated with vehicle (1% DMSO/0.9% saline), 0.1 mg/kg (mpk) ISRIB, or 0.25 mpk ISRIB ($N = 15$ per group) 

**Figure 2 (…continued)**
immediately after completion of a MWM training session (1 trial per day). No differences in learning rates to find the hidden platform, in either latency or distance measures, were observed between mice treated with vehicle or ISRIB (linear mixed effects regression model, latency Trial*ISRIB 0.1 mpk = −1.2 sec/day (5–95% CI [−10.5–8.1] sec/day), latency Trial*ISRIB 0.25 mpk = −0.9 sec/day (5–95% CI [−10.1–8.3] sec/day), distance Trial*ISRIB 0.1 mpk = −35 cm/day (5–95% CI [−207–136] cm/day), distance Trial*ISRIB 0.25 mpk = −36 cm/day (5–95% CI [−207–135] cm/day)). (B) Mice were tested in a probe trial 24 hours after completion of the last training trial. No significant improvement in memory for the location of the hidden platform was observed in multiple outcome measures, including percentage of time spent in the target quadrant (one sample $t$ test against 25% (random chance, dotted line): vehicle $p = 0.03$, ISRIB 0.1 mpk $p = 0.23$, ISRIB 0.25 mpk $p = 0.09$; one-way ANOVA, $F_{(2,42)} = 0.45$, $p = 0.64$) latency to first cross the target platform location (one-way ANOVA, $F_{(2,39)} = 1.18$, $p = 0.32$), and the number of platform crossings (one-way ANOVA, $F_{(2,42)} = 0.93$, $p = 0.40$), although numerically the values trended in the expected direction in the latency to target and crossing frequency measures in the 0.25 mpk ISRIB group. (C) Mice were tested in a probe trial 72 hours after completion of the last training trial. Mice treated with 0.25 mpk ISRIB spent more time in the target quadrant compared to chance (one sample $t$ test against 25%: vehicle $p = 0.26$, ISRIB 0.1 mpk $p = 0.85$, ISRIB 0.25 mpk $p = 0.009$; one-way ANOVA, $F_{(2,42)} = 2.57$, $p = 0.09$). There was no significant improvement in latency to target (one-way ANOVA, $F_{(2,32)} = 1.50$, $p = 0.24$) or crossing frequency (one-way ANOVA, $F_{(2,42)} = 0.38$, $p = 0.69$) outcome measures with ISRIB, although the same numerical trends in latency to target and, to a lesser extent, crossing frequency were observed as in the 24-hour probe trial. $*p < 0.05$, $**p < 0.01$.

**Table 1 ISRIB pharmacokinetic data.** mpk = mg/kg. The IC$_{50}$ of ISRIB is 5nM. ISRIB demonstrates excellent blood-brain-barrier penetration (*Sidrauski et al., 2013*; *Halliday et al., 2015*).

| Experiment | Mouse | ISRIB treatment (vehicle) | Time post-administration | Plasma concentration (ng/mL) | Plasma concentration (nM) |
|---|---|---|---|---|---|
| Morris water maze #1 | 1 | 0.25 mpk (1% DMSO/saline) | 2.5 hours | 65.3 | 145 |
| | 2 | 0.25 mpk (1% DMSO/saline) | 2.5 hours | 71.9 | 159 |
| Morris water maze #2 | 1 | 0.25 mpk (1% DMSO/saline) | 2.5 hours | 66.6 | 148 |
| | 2 | 0.25 mpk (1% DMSO/saline) | 2.5 hours | 65.6 | 145 |
| | 3 | 2.5 mpk (DMSO/PEG) | 2.5 hours | 657 | 1456 |
| | 4 | 2.5 mpk (DMSO/PEG) | 2.5 hours | 348 | 771 |
| | 5 | 2.5 mpk (DMSO/PEG) | 2.5 hours | 286 | 634 |

(one sample $t$ test against 25%: vehicle $p = 0.26$, ISRIB 0.1 mpk $p = 0.85$, ISRIB 0.25 mpk $p = 0.009$) (Fig. 2C). There was also a general numerical trend towards improvement in all three memory outcome measures with 0.25 mpk ISRIB compared to control (Fig. 2C). We verified drug exposure on day 6 of the MWM by injecting 2 satellite mice with 0.25 mpk ISRIB and measuring plasma levels 2.5 hours post-injection. Both satellite mice showed plasma levels above the IC$_{50}$ (5nM) for ISRIB (Table 1) (*Sidrauski et al., 2013*). ISRIB is not significantly excluded from the brain by the blood–brain-barrier, so plasma ISRIB levels roughly approximate compound levels in the brain (*Sidrauski et al., 2013*; *Halliday et al., 2015*). Thus, while we did not observe a benefit in training measures with ISRIB treatment in this pilot study, a number of outcome measures in the probe trials suggested a general statistically insignificant trend towards improvement in long-term memory with ISRIB treatment, especially at the 0.25 mpk dose.

## ISRIB did not affect memory in contextual fear conditioning

Given our observations in the Morris water maze probe trials, we decided to use the same mice to test whether contextual fear memory, which is largely dependent upon the amygdala and hippocampus, could be enhanced with ISRIB treatment. For this assay, we used a different vehicle formulation for ISRIB in order to increase the amount of compound that could be delivered by the i.p. route of administration. We found that a minimal volume of 50% DMSO/ 50% polyethyleneglycol (PEG) was sufficient to dissolve and deliver the hydrophobic ISRIB compound at a 10-fold greater dose (2.5 mpk) than what could be achieved with 1% DMSO/saline during the pilot MWM experiment. Nine days after completion of the MWM 72-hr probe trial, mice were placed in fear conditioning chambers and observed for freezing behavior prior to and after delivery of a mild electrical foot shock. Prior to receiving the shock, the control and planned treatment group both showed very low freezing behavior. After delivery of the shock, both groups demonstrated increased freezing behavior, with no significant difference between control and planned treatment groups (Mann–Whitney test, $p = 0.18$) (Fig. 3A). After removal from the chambers, the mice were treated with 2.5 mpk ISRIB with a single i.p. injection and placed back into their home cages. Twenty-four hours later, the mice were placed into the chamber (context) and their freezing behavior quantified. Over the first 4 min, the second 4 min, or the total 8 min of observation, we did not observe a significant difference in freezing behavior between vehicle- and ISRIB-treated groups (first 4 min unpaired $t$ test, $p = 0.38$; second 4 min Mann–Whitney test, $p = 0.24$; total 8 min Mann–Whitney test, $p = 0.31$). (Fig. 3B). Thus, we did not observe an enhancement of fear memory with ISRIB using this testing paradigm.

## ISRIB failed to rescue spatial learning and memory deficits in hAPP-J20 mice

Given the potential trend toward improvement in memory in the MWM with ISRIB at 0.25 mpk, and the ability to deliver greater amounts of the compound using the DMSO/PEG vehicle, we proceeded to test whether ISRIB might improve learning and memory in the hAPP-J20 (J20) model of Alzheimer's disease. A cohort consisting of nontransgenic (NTG) and J20 mice aged 6–7 months were used to test enhancement of spatial learning and memory in the MWM with two separate doses of ISRIB: 0.25 mpk in 1% DMSO/saline, and 2.5 mpk in DMSO/PEG. Because we were unsure how the DMSO/PEG vehicle used to deliver the higher ISRIB dose may affect the mice, we controlled for both vehicle formulations. Each control and treatment group consisted of approximately 12–15 mice. To increase memory for the location of the platform compared to the pilot MWM study, we performed 5 additional training sessions, with 2 trials per day on the last 2 days. Mice were treated with vehicle (1% DMSO/saline or DMSO/PEG) or ISRIB at 0.25 mpk or 2.5 mpk i.p. immediately after completion of each training session. Probe trials were performed at 24 and 72 hours after the last training session to test spatial memory. While we observed a significant learning deficit in J20 compared to NTG mice (Fig. S1), we did not observe an improvement in learning during the training trials with ISRIB treatment at either dose in NTG or J20 mice (linear mixed effects regression models,

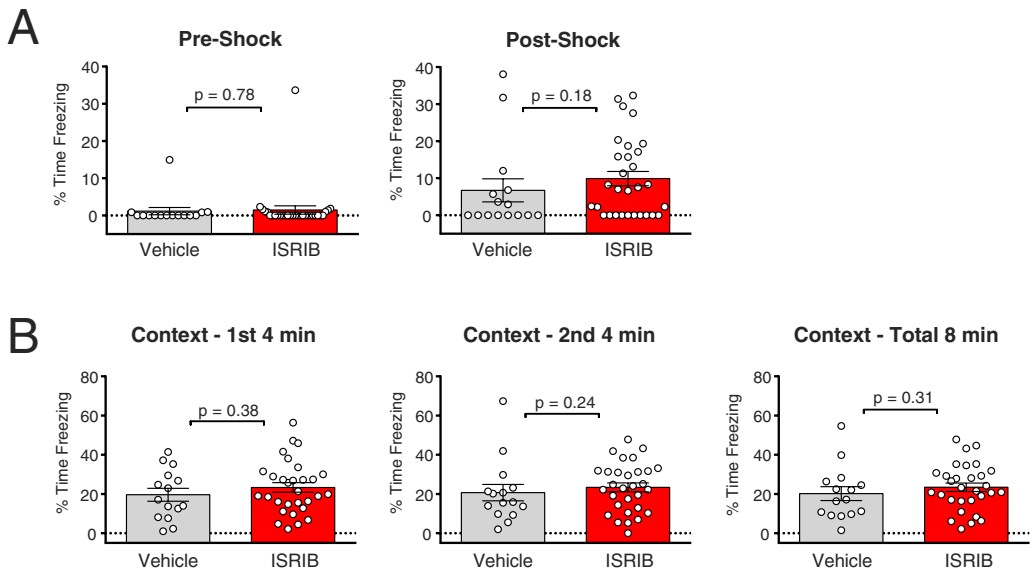

**Figure 3  ISRIB does not show a beneficial effect on fear memory in a contextual fear conditioning paradigm.** (A, B) Mice that had been tested previously in the MWM were tested in fear conditioning. (A) Mice were placed in the chamber and allowed to explore for 2 minutes prior to delivery of a mild foot shock, after which their freezing behavior was quantified for 1 minute. They were then removed from the chamber and immediately treated with either vehicle (1:1 DMSO:PEG$_{400}$, $N = 15$) or 2.5 mpk ISRIB ($N = 30$). The mice showed no differences in freezing rates between treatment groups prior to and after receiving the foot shock (Mann–Whitney test). (B) 24 hours later the mice were placed back in the same chamber (context) and their freezing behavior was quantified over the course of 8 minutes. No difference in the amount of freezing was observed between vehicle- and ISRIB-treated groups during the first 4 minutes, the second 4 minutes, or the total 8 minutes of observation (first 4 minutes unpaired $t$ test, second 4 minutes Mann-Whitney test, total 8 minutes Mann-Whitney test).

non-significant 5–95% confidence intervals) (Figs. 4A–4D). We also did not observe an improvement in 3 separate measures of spatial memory during the 24 hour probe trial in NTG or J20 mice at either dose (Figs. 4E–4G; see figure legend for statistical details). When we tested the mice again at 72 hours in the probe trial (Figs. 4H–4J), we did not observe significantly positive treatment effects (see figure legend for statistical details). To increase our power to detect an ISRIB treatment effect, we pooled the vehicle control groups in NTG and J20 mice and reanalyzed the data (Fig. S2). After pooling control groups, we did not detect an ISRIB treatment effect (24 hour time in target quadrant two-way ANOVA: treatment $F_{(2,87)} = 0.42$, $p = 0.66$; genotype $F_{(1,87)} = 25.10$, $p < 0.0001$; treatment × genotype $F_{(2,87)} = 0.06$, $p = 0.95$. 72 hour time in target quadrant two-way ANOVA: treatment $F_{(2,87)} = 0.61$, $p = 0.55$; genotype $F_{(1,87)} = 11.15$, $p = 0.001$; treatment × genotype $F_{(2,87)} = 1.73$, $p = 0.18$. See Fig. S2 for additional outcome measures). To verify that we had delivered the ISRIB compound to the mice, we tested plasma levels of ISRIB in a satellite cohort of mice 2.5 hours after i.p. injection during trial 7. Plasma ISRIB levels were above the IC$_{50}$ (5nM) in all mice (Table 1). We also tested whether ISRIB was chemically stable in our DMSO stock solution used to prepare the treatment solutions during MWM testing. Liquid chromatography-mass spectrometry (LC-MS) analysis of the stock solution at $t = 0$ and $t = 3$ weeks showed no change in retention time or mass of the

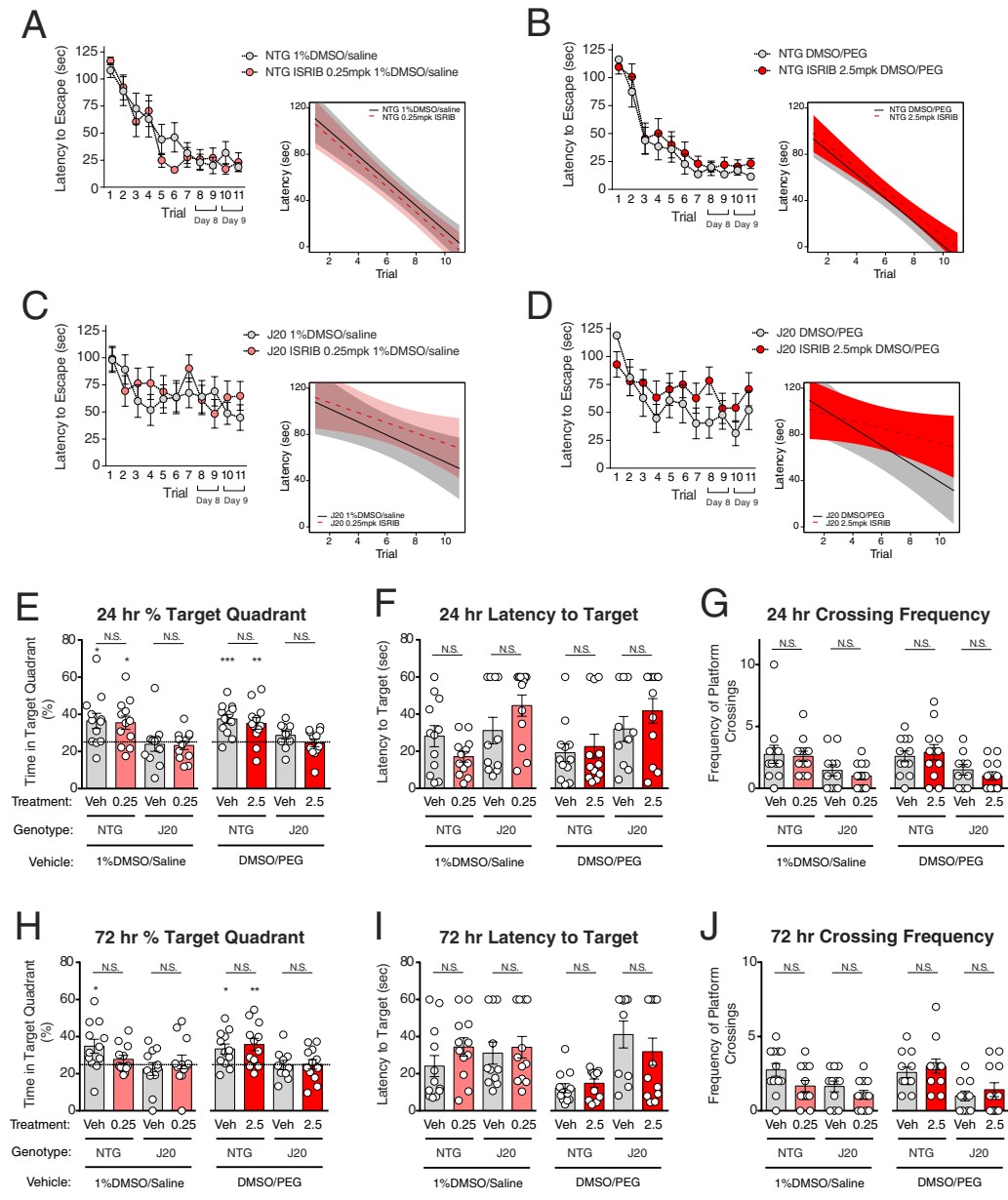

**Figure 4  ISRIB does not rescue spatial learning and memory deficits in hAPP-J20 mice in the Morris water maze.** (A–J) A separate cohort of 6–7 month-old nontransgenic (NTG) and hAPP-J20 (J20) mice treated with either vehicle or ISRIB were tested in the MWM. (A–D) Mice were trained in the MWM over 11 trials. Mice were treated with vehicle (1% DMSO/0.9% saline or 50% DMSO/50% PEG) or ISRIB (0.25 mpk or 2.5 mpk) immediately after each training session. On days 8 and 9, mice received two training sessions per day, and were injected immediately after the second session. (A) NTG mice were treated with vehicle (1% DMSO/saline, $N = 12$) or ISRIB (0.25 mpk in 1% DMSO/saline, $N = 12$). (B) NTG mice were treated with vehicle (50% DMSO/50% PEG, $N = 12$) or ISRIB (2.5 mpk in 50% DMSO/50% PEG, $N = 12$). (C) J20 mice were treated with vehicle (1% DMSO/saline, $N = 11$) or ISRIB (0.25 mpk in 1% DMSO/saline, $N = 12$). (D) J20 mice were treated with vehicle (50% DMSO/50% PEG, $N = 10$) or ISRIB (2.5 mpk in 50% DMSO/50% PEG, $N = 12$). (Insets) Linear mixed effects regression 

![PeerJ]

**Figure 4 (. . . continued)**
models were used to analyze differences in learning rates between treatment and vehicle control groups. No significant differences in learning rates were observed in NTG mice or J20 mice with ISRIB treatment at either dose (linear mixed effects regression model, latency NTG Trial*ISRIB 0.25 mpk = −0.1 sec/day (5–95% CI [−3.9–3.7] sec/day), latency NTG Trial*ISRIB 2.5 mpk = 0.3 sec/day (5–95% CI [−3.0–3.8] sec/day), latency J20 Trial*ISRIB 0.25 mpk = −1.4 sec/day (5–95% CI [−4.2–6.9] sec/day), latency J20 Trial*ISRIB 2.5 mpk = 4.6 sec/day (5–95% CI [−0.8–9.9] sec/day)). Shaded areas indicate 5–95% confidence intervals. (E–G) A probe trial was performed 24 hours after completion of the last training trial. (E) NTG mice spent more time in the target quadrant than would be expected by chance, while J20 mice did not (one sample $t$ test against 25% (dotted line): NTG 1% DMSO/saline vehicle $p = 0.02$, NTG IS-RIB 0.25 mpk $p = 0.01$, J20 1% DMSO/saline vehicle $p = 0.76$, J20 ISRIB 0.25 mpk $p = 0.45$, NTG DMSO/PEG vehicle $p = 0.0003$, NTG ISRIB 2.5 mpk $p = 0.008$, J20 DMSO/PEG vehicle $p = 0.11$, J20 ISRIB 2.5 mpk $p = 0.87$). Treatment with ISRIB at 0.25 mpk or 2.5 mpk did not increase the percentage of time spent in the target quadrant in either group (two-way ANOVA for ISRIB 0.25 mpk: treatment $F_{(1,43)} = 0.054$, $p = 0.83$; genotype $F_{(1,43)} = 12.88$, $p = 0.0008$; treatment $\times$ genotype $F_{(1,43)} = 0.003$, $p = 0.96$. Two-way ANOVA for ISRIB 2.5 mpk: treatment $F_{(1,42)} = 1.66$, $p = 0.20$; genotype $F_{(1,42)} = 14.68$, $p = 0.0004$; treatment $\times$ genotype $F_{(1,42)} = 0.11$, $p = 0.74$). (F) Treatment with ISRIB did not reduce the latency to first cross the target platform location (two-way ANOVA for ISRIB 0.25 mpk: treatment $F_{(1,43)} = 0.04$, $p = 0.83$; genotype $F_{(1,43)} = 7.75$, $p = 0.008$; treatment $\times$ genotype $F_{(1,43)} = 4.98$, $p = 0.03$. Two-way ANOVA for ISRIB 2.5 mpk: treatment $F_{(1,42)} = 1.12$, $p = 0.30$; genotype $F_{(1,42)} = 6.62$, $p = 0.01$; treatment $\times$ genotype $F_{(1,42)} = 0.26$, $p = 0.61$). (G) Treatment with ISRIB did not increase the number of platform crossings (two-way ANOVA for ISRIB 0.25 mpk: treatment $F_{(1,43)} = 0.38$, $p = 0.54$; genotype $F_{(1,43)} = 8.17$, $p = 0.007$; treatment $\times$ genotype $F_{(1,43)} = 0.08$, $p = 0.78$. Two-way ANOVA for ISRIB 2.5 mpk: treatment $F_{(1,42)} = 0.03$, $p = 0.86$; genotype $F_{(1,42)} = 9.97$, $p = 0.003$; treatment $\times$ genotype $F_{(1,42)} = 0.77$, $p = 0.39$). (H–J) A probe trial was performed 72 hours after completion of the last training trial. (H) NTG mice continued to spend more time in the target quadrant compared to chance, except for the group treated with 0.25 mpk ISRIB (one sample $t$ test against 25% (dotted line): NTG 1% DMSO/saline vehicle $p = 0.02$, NTG ISRIB 0.25 mpk $p = 0.23$, J20 1% DMSO/saline vehicle $p = 0.48$, J20 ISRIB 0.25 mpk $p = 0.73$, NTG DMSO/PEG vehicle $p = 0.01$, NTG ISRIB 2.5 mpk $p = 0.008$, J20 DMSO/PEG vehicle $p = 0.88$, J20 ISRIB 2.5 mpk $p = 0.99$). No significant treatment effect was observed in either group (two-way ANOVA for ISRIB 0.25 mpk: treatment $F_{(1,43)} = 0.23$, $p = 0.63$; genotype $F_{(1,43)} = 4.22$, $p = 0.04$; treatment $\times$ genotype $F_{(1,43)} = 2.66$, $p = 0.11$. Two-way ANOVA for ISRIB 2.5 mpk: treatment $F_{(1,42)} = 0.24$, $p = 0.62$; genotype $F_{(1,42)} = 11.85$, $p = 0.001$; treatment $\times$ genotype $F_{(1,42)} = 0.13$, $p = 0.72$). (I) Treatment with ISRIB did not reduce the latency to first cross the target platform location (two-way ANOVA for IS-RIB 0.25 mpk: treatment $F_{(1,43)} = 1.44$, $p = 0.24$; genotype $F_{(1,43)} = 0.37$, $p = 0.54$; treatment $\times$ genotype $F_{(1,43)} = 0.40$, $p = 0.53$. Two-way ANOVA for ISRIB 2.5 mpk: treatment $F_{(1,42)} = 0.39$, $p = 0.53$; genotype $F_{(1,42)} = 19.00$, $p < 0.0001$; treatment $\times$ genotype $F_{(1,42)} = 1.28$, $p = 0.26$). (J) Treatment with ISRIB did not increase the number of platform crossings (two-way ANOVA for ISRIB 0.25 mpk: treatment $F_{(1,43)} = 4.94$, $p = 0.03$, Sidak's post-hoc comparisons NTG vehicle vs. ISRIB CI [−0.11–2.28], J20 vehicle vs. ISRIB CI [−0.67–1.77]; genotype $F_{(1,43)} = 5.31$, $p = 0.03$; treatment $\times$ genotype $F_{(1,43)} = 0.52$, $p = 0.48$. Two-way ANOVA for ISRIB 2.5 mpk: treatment $F_{(1,42)} = 0.97$, $p = 0.33$; genotype $F_{(1,42)} = 14.02$, $p = 0.0005$; treatment $\times$ genotype $F_{(1,42)} = 0.0$, $p > 0.99$). $*p < 0.05$, $**p < 0.01$, $***p < 0.001$. CI, confidence interval; N.S., not significant; Veh, vehicle; 0.25, ISRIB 0.25 mg/kg; 2.5, ISRIB 2.5 mg/kg; DMSO, dimethyl sulfoxide; PEG, polyethylene glycol.

ISRIB compound (data not shown, available upon request). Thus, ISRIB did not reverse spatial learning or memory deficits in J20, nor did it improve memory in NTG mice in this MWM experiment despite adequate drug exposure.

## DISCUSSION

In this preclinical drug study we tested whether ISRIB would rescue the learning and memory deficits observed in the J20 mouse model of Alzheimer's disease. We also tested whether it would enhance spatial learning and memory, as well as fear memory, in

nontransgenic mice. We did not observe a significant rescue of spatial learning or memory in 6–7 month-old J20 mice with ISRIB. In nontransgenic mice, we observed a general trend towards enhanced long-term memory at the 0.25 mpk dose in a pilot MWM cohort, but we were unable to repeat this finding in a second cohort. We also did not find an enhancement of fear memory at 24 hours in 6–7 month-old nontransgenic mice. Although we did not have a marker of target engagement for these experiments, similar to previous studies on memory enhancement with ISRIB (*Sidrauski et al., 2013*; *Di Prisco et al., 2014*), we measured plasma levels of ISRIB in two satellite cohorts of mice during the MWM experiments and verified excellent drug exposure at both the 0.25 mpk and 2.5 mpk doses. ISRIB readily crosses the blood–brain-barrier (*Sidrauski et al., 2013*; *Halliday et al., 2015*).

ISRIB has been shown to extend survival in disease models that show elevated ISR activity, such as prion disease mouse models (*Halliday et al., 2015*). In a prion disease model, treatment with ISRIB reduced inhibition of protein translation as measured by the levels of ATF4, a marker of ISR activity, and extended survival by approximately 1 week. This beneficial effect on survival was achieved by chronic daily i.p. administration of 0.25 mpk ISRIB over the course of five to seven weeks (*Halliday et al., 2015*). In contrast, ISRIB's effects on learning and memory have been demonstrated after a single i.p. injection in wildtype rodents (*Sidrauski et al., 2013*; *Di Prisco et al., 2014*), suggesting that chronic administration of ISRIB is not required for cognitive enhancement, nor presumably for rescue of cognitive deficits via the same mechanism, at least in disease models where the ISR is not elevated such as in the J20 mouse model. Both the beneficial survival effect in prion-infected mice and the beneficial effects on cognition in NTG mice with ISRIB treatment are thought to relate to the compound's effects on protein translation. It is possible that chronic dosing of ISRIB, most easily performed through a food formulation, would also lead to an improvement in learning and memory. This would be an interesting future direction for research on ISRIB. Given that protein translation rates are highly regulated, it seems quite possible that increasing total protein translation throughout an organism for an extended period of time could in fact be detrimental to cognition, either directly or through a secondary toxicity mechanism. It is also unclear whether chronic ISRIB administration would allow homeostatic compensation to occur in the protein translation system prior to the learning test, thereby preventing the desired enhancement on learning and memory. Answering such questions would be highly facilitated by a marker of ISRIB target engagement that is directly related to the underlying mechanism(s) by which it is thought to enhance cognition. One such mechanism that has been proposed is modulation of long term depression (LTD) (*Di Prisco et al., 2014*). Object-place learning has been shown to be dependent upon proper induction of hippocampal LTD, which appears to require a transcriptional program that is induced by eIF2$\alpha$ phosphorylation (*Di Prisco et al., 2014*). Acute treatment with ISRIB interfered with proper object-place learning by preventing activation of this transcriptional program and induction of LTD (*Di Prisco et al., 2014*). Interestingly, A$\beta$ has been shown to cause elevated LTD in rodent hippocampal slice culture (*Shankar et al., 2008*; *Luscher & Huber, 2010*; *Pozueta, Lefort & Shelanski, 2013*), which may be one mechanism by which hAPP overexpression contributes to learning and memory deficits observed in AD mouse models. Performance in these behavioral assays can

also be influenced by non-neuronal cell types such as astrocytes (*Orr et al., 2015*), which are pathologically activated in the presence of A$\beta$ plaques (*Serrano-Pozo et al., 2016*) and are also likely affected by treatment with ISRIB. Cell-type specific biochemical markers, rather than electrophysiologic markers, of target engagement that relate to performance in these behavioral tasks would be helpful for the study of ISRIB's effects on cognition in wildtype and disease models.

In addition to the lack of effects observed in the J20 model, we were unable to replicate the memory-enhancing effects with ISRIB treatment in nontransgenic mice. There are a number of possibilities that might explain why we were unable to see an effect on memory in nontransgenic mice. The first possibility is that our training paradigm in the MWM was not weak enough to allow a benefit to emerge in ISRIB-treated mice. While this possibility may explain our inability to see an effect during training in the pilot MWM #1 and MWM #2 given that all NTG mice learned the location of the hidden platform at equivalent rates, it is unlikely to explain our inability to see an effect in the probe trials, as the control mice in MWM #1 showed rather weak memory for the location of the platform at 24 and 72 hours. When we increased training in MWM #2 with an additional 5 trials, we noted improved memory for the location of the platform in NTG mice at 24 and 72 hours compared to mice tested in MWM #1, as expected. While we again noted a general trend towards improvement in latency to target at the 0.25 mpk dose similar to the observation in the pilot MWM #1, we did not find a significant improvement in memory in NTG mice in MWM #2. Therefore, while a weaker training protocol may make it more likely for differences to be observed in learning rates with ISRIB treatment, it is unlikely to lead to better separation between groups in the probe trial. A second possibility is that our study was insufficiently powered to see a beneficial effect on learning and memory in NTG mice. Given the general trends observed in MWM #1 and the relatively small effect size, a study that is much more highly powered might be able to show significance. We note that each experimental group in this study contained approximately 12–15 mice, for a total of >100 mice (including satellite mice) for MWM #2. The size of each experimental group is equivalent to or larger than those used in previous behavioral studies with ISRIB (*Sidrauski et al., 2013*; *Di Prisco et al., 2014*). To increase power in a future MWM experiment, the number of treatment groups would need to be reduced, and/or separate experiments would need to be analyzed in a pooled manner. A third (and related) possibility is that the MWM is a relatively insensitive test of spatial learning and memory. Newer behavioral assays of spatial learning and memory, such as the interactive place avoidance test, might better detect an enhancement of learning and memory with ISRIB treatment (*Cimadevilla, Fenton & Bures, 2001*). A final possibility is that the memory-enhancing effect of ISRIB is age-dependent. In our study we tested 6–7 month-old mice, whereas the prior study that demonstrated memory enhancement with ISRIB used 2 month-old mice.

One potential reason why we were unable to observe an enhancement of fear memory with ISRIB may have been the sequencing of the behavioral tests performed. Contextual fear conditioning was performed in the same cohort of mice after they had been exposed to the MWM. This may have explained their rather low baseline freezing rates prior to receiving a foot shock, as desensitization may have occurred with prior exposure to the pool

of water in the MWM. Future tests of fear memory enhancement with ISRIB treatment would likely benefit from using a naïve cohort of mice.

In conclusion, we did not find a significant beneficial effect of ISRIB treatment in hAPP-J20 mice or NTG mice. Future preclinical studies with ISRIB on memory enhancement in wildtype animals and disease models would benefit from robust markers of target engagement in the brain that are directly related to the molecular mechanism(s) by which memory enhancement is thought to occur.

## ACKNOWLEDGEMENTS

The authors thank Gui-qiu Yu, Xin Wang, and Wei-kun Guo for mouse colony maintenance, Shaun Fontaine for performing LC-MS analysis, and Mariel Finucane for advice on statistical model construction and analysis. The authors thank Carmela Sidrauski, Pascal Sanchez, Kaitlyn Ho, and the Gladstone Behavioral Core staff for helpful discussions, and Lennart Mucke for reviewing the manuscript. The authors also thank Monica Dela Cruz, Amy Cheung, and Courtney Dickerson for administrative assistance.

### Funding
EJ was supported by an NINDS R25 fellowship (R25NS070680). The funders had no role in study design, data collection and analysis, decision to publish, or preparation of the manuscript.

### Grant Disclosures
The following grant information was disclosed by the authors:
NINDS R25 fellowship: R25NS070680.

### Competing Interests
The authors declare there are no competing interests.

### Author Contributions
- Erik C.B. Johnson conceived and designed the experiments, performed the experiments, analyzed the data, wrote the paper, prepared figures and/or tables, reviewed drafts of the paper.
- Jing Kang performed the experiments, reviewed drafts of the paper.

### Animal Ethics
The following information was supplied relating to ethical approvals (i.e., approving body and any reference numbers):

All animal experiments were approved by the Institutional Animal Care and Use Committee at University of California-San Francisco, under protocol AN105527-03.

### Data Availability
The raw data has been supplied as Supplementary File.

## Supplemental Information

Supplemental information for this article can be found online at http://dx.doi.org/10.7717/peerj.2565#supplemental-information.

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
