# Peer review of "A small molecule targeting protein translation does not rescue spatial learning and memory deficits in the hAPP-J20 mouse model of Alzheimer’s disease"

_PeerJ, doi:10.7717/peerj.2565_

## Round 0.1 · original submission · Major Revisions

The reviewers and I agree that the manuscript is well written, experiments are well conducted on the whole, and the figures present the data well. However, I would highly recommend that the following changes are made before you consider a resubmission of your paper.

Introduction - this is a very brief introduction, and requires more elaboration in parts. For instance, it could benefit from a more detailed description of the previous studies that have reported enhanced learning and memory following ISRIB administration, to put the present study into context. Has the present study attempted to use the same ISRIB administration protocol/ behavioural procedures as the previous studies?I concur with Reviewer 2 on the need for a more clearly defined rationale for the use of the J20 mouse model, especially as the use of the J20 model sets your paper apart from the other studies.

Methods - As referred to by two of the reviewers, please provide a rationale for the dosing protocol that was used in the study, as well as address other quires raised by all the reviewers.

Results - I agree with Reviewer 3 that the statistical reporting of the results is rather confusing, making it difficult to assess the validity of the findings. Please follow Reviewer 3's suggestions for improvement. In addition, please explicitly state whether the water maze performance of the J20 mice was significantly impaired compared to the nontransgenic group performance.

Discussion - Two of the reviewers raised valid points regarding 1) the validity of the use of the J20 model to test the efficacy of ISRIB, given that ISR levels were not raised in these mice, and 2) the choice of dosing protocol. These points need to be addressed. In addition, Reviewer 3 has some very good suggestions about strengthening the discussion.

Reviewer 1 ·

Basic reporting

In this paper, the authors have tried to replicate the behavioural results from Sidrauski et al., (2013) which describes the discovery of ISRIB and improvements in learning and memory after administration of ISRIB in rodents. They have also tested ISRIB in the hAPP-J20 mouse model of Alzheimer’s disease. The authors do not observe any change in learning and memory in wild type and hAPP-J20 mice, as measured by the Morris water maze and fear conditioning behavioural tests. The paper is well written, and is well placed within the context of the field. The figures are well made and clearly depict the data presented.

Experimental design

The water maze and fear conditioning tests have been performed in a suitable manner and to a high standard, and the methods are detailed and clear enough to be replicated. However, there are concerns around the ISRIB dosing in this study. ISRIB is only dosed immediately after the training phases in the behavioural tests. It does not appear that any further dose was administered between the training and test phases, which could be as much as 72 hours apart. After this time period, most/all of the administered ISRIB is likely to be excreted or metabolized. Daily dosing of ISRIB, for a period before the training, during the training, and up to the time of the test experiments would provide a much better test of ISRIB’s effects on learning and memory.

Also, as no LC/MS was performed on brain tissue, the authors cannot be certain that ISRIB is effectively entering the brain. Sidrauski et al., (2013) and Halliday et al., (2015) both show a good correlation between plasma and brain ISRIB levels, but due to the negative results presented in this paper, the authors need to be sure that suitable levels of ISRIB are present in the brain.

ISRIB is highly insoluble, and absorbed levels can vary greatly between mice receiving the same dose. If plasma levels are to be used as a correlate of brain levels, plasma needs to be examined from the mice performing the behavioural studies, rather than in satellite mice, where levels may be significantly different.

Also, testing ISRIB in hAPP-J20 mice that do not show any ISR activation is not the best model to test the effectiveness of the compound.

Validity of the findings

Publishing negative results and replicating the findings of others are both important and worthy of publication. However, the authors cannot be sure if this is a true negative result or if they were unable to detect any effects of ISRIB treatment due to inconsistencies in the dosing or the choice of model used. Therefore, until a clear marker of target engagement is found, publishing this paper may confuse the readership to the true effectiveness of ISRIB.

·

Basic reporting

The Introduction insufficiently delineates the rationale for assessing J20 mice in this context. Authors should clarify the rationale for choosing J20 mice, specifically, as well as the potential relevance of the mechanism of action of ISRIB to Alzheimer's disease. This is particularly relevant considering the data in Fig 1 suggest that ISR is not elevated in J20 mice.

Experimental design

Better rationale for drug administration schedule should be provided. Given the putative mechanism of action of ISRIB, why was chronic administration not considered? This could be suggested in the discussion.

Line 116: What is a "near infrared fear conditioning chamber"?

Validity of the findings

Thorough data sets are provided, as well as many relevant control experiments.
Authors question their own power. Can a power analysis be provided? I doubt this study actually had insufficient power.

Reviewer 3 ·

Basic reporting

Fine. I have no specific comments

Experimental design

For the behavioural work, the study closely follows the design of previous work by Sidrauski et al. that did find a significant effect in normal mice in the water maze and in contextual fear conditioning. However, unfortunately for the authors of the current paper, there isn’t a lot of detail in Sidrauski paper so we are left wondering whether there were subtle differences in the procedures of the two studies that were crucial for determining the differences in the results.

It is not clear whether the location of the platform in the water maze task was counterbalanced across mice. It could be possible that unlearnt preferences for particular locations affect performance, which may have reduced the chances of finding a significant effect. However, given that we don’t know whether the platform location was counterbalanced in the Sidrauski paper, an effect of ISIRB on unlearnt spatial preferences in their study cannot be ruled out.

A potential issue in the fear conditioning study that is relevant for both the current study and the Sidrauski study is that there is no control procedure. Mice freeze more after having a shock, but it is not clear whether they freeze because exposure to a shock is sufficient to increase freezing in any situation (i.e. a kind of pseudoconditioning effect) or whether they have learnt that the context signals fear. Assessment of freezing in an equally familiar context that hasn’t been paired with shock would address this question.

It is not clear whether the control mice in the J20 study are littermate controls.

Validity of the findings

The results are clear. However, the stats seem overly complicated and at times, confusing. Some of the near trends were not obvious to me. For example, it is stated that there was a trend for the 24 hr latency to target test to be affect by the drug, but the p value = 0.32. I think the suggestion is that the results are numerically going in the right direction, but they are not significant. I think the text should be reworded to avoid confusion.

I would like to see the actual statistic (F values, t values etc.) and degrees of freedom for each test.

I think it is misleading to state whether individual groups were above chance or not when there was no effect of group in the ANOVA. If there is no effect of group I think it is only useful to state whether all animals, independent of group, were above chance.

When tests are conducted at 24 hours and at 72 hours, it may be simpler to combine the analysis as a repeated measure. In addition to increasing the power of the analysis, it also allows extinction of learning to be gauged. It could be useful.

Also, why have separate analyses for the different doses of ISIRB? Either the control groups could be pooled, if they don’t differ, and then the two doses could be compared to one control group. Alternatively, given the different control groups, combine in a 2(control, drug) by 2(0.25/DMSO, 2.5/DMSO-PEG) design.

Additional comments

I think the results that are most striking are the lack of effect of ISIRB on the deficits found in the hAPP-J20 Alzheimer’s mice. I think the discussion should be more heavily weighted in discussing the implications of this null result given the original rationale for the work. It is harder to know what to make of the lack of effect in normal mice given the results of the Sidrauski study. If I have understood correctly, the work in normal mice is a direct replication of the Sidrauski study. Although, there may be reasons why the current study failed to see effects (such as experiment order), at the least the results suggest that the effects reported by Sidrauski may not be as big as first thought. This could be stated more strongly, if the authors felt that it was appropriate.

---

## Round 0.2 · Minor Revisions

Thank you for your thorough response to all the comments and suggestions for revision. The manuscript is much improved, but I have one outstanding concern about the format of the Results section. I recommend that the authors bring some statistical reporting back into the main text. This information can be kept minimal, but it would be useful for the readers to see the significance (and non-significance) levels (F and p values) in brackets as they read the results section.

Reviewer 1 ·

Basic reporting

No additional comments

Experimental design

No additional comments

Validity of the findings

No additional comments

Additional comments

The authors have improved the paper and addressed the majority of the points raised by the reviewers and editor. My only concern still is that due to the positive results using ISRIB by other groups, the compound is likely to have observable effects on learning and memory that the author's have not observed. However, under the conditions and using the mice reported in this paper, I agree the work is performed well enough to be valid. ISRIB is a new compound with a small number of papers published on it, so the full range of ISIRB's effects (or lack thereof) should be part of the scientific literature.

·

Basic reporting

No comments.

Experimental design

No comments.

Validity of the findings

No comments.

Additional comments

The authors have adequately addressed my comments.

---

## Round 0.3 · accepted · Accept

All concerns have now been addressed.